# Evaluation of the Main Macro-, Micro- and Trace Elements Found in *Fallopia japonica* Plants and Their Traceability in Its Honey: A Case Study from the Northwestern and Western Part of Romania

**DOI:** 10.3390/plants13030428

**Published:** 2024-01-31

**Authors:** Alexandra-Antonia Cucu, Claudia Pașca, Alexandru-Bogdan Cucu, Adela Ramona Moise, Otilia Bobiş, Ștefan Dezsi, Anamaria Blaga Petrean, Daniel Severus Dezmirean

**Affiliations:** 1Faculty of Animal Science and Biotechnology, University of Animal Sciences and Veterinary Medicine Cluj-Napoca, 3-5 Calea Manastur St., 400372 Cluj-Napoca, Romania; antonia.cucu@usamvcluj.ro (A.-A.C.); adela.moise@usamvcluj.ro (A.R.M.); 2National Institute for Research and Development in Forestry (INCDS) “Marin Drăcea”, 400202 Braşov, Romania; alexandru.cucu@icas.ro; 3Faculty of Geography, Babeş-Bolyai University, 400084 Cluj-Napoca, Romania; stefan.dezsi@ubbcluj.ro; 4Department of Animal Production and Food Safety, Faculty of Veterinary Medicine, University of Agricultural Sciences and Veterinary Medicine Cluj-Napoca, 3-5 Calea Manastur St., 400372 Cluj-Napoca, Romania; anamaria.petrean@usamvcluj.ro

**Keywords:** *Fallopia japonica*, invasive plant, minerals, knotweed honey, trace elements, beekeeping, toxic element, polluted sites

## Abstract

*Fallopia japonica* (Japanese knotweed, *Reynoutria japonica* or *Polygonum cuspidatum)* is considered an extremely invasive plant worldwide and a bioindicator of heavy metals. Yet, its potential as a crop for honeybees is still underevaluated. This study employs atomic absorption spectrometry to quantitatively analyze the concentration of macro-elements, namely, calcium (Ca), potassium (K) and magnesium (Mg); micro-elements, such as copper (Cu), iron (Fe), manganese (Mn) and selenium (Se); and trace elements, i.e., cadmium (Cd), chromium (Cr), nickel (Ni) and lead (Pb) in different anatomic parts of *Fallopia japonica* (FJ) plants (roots, rhizomes, stems, leaves) and their traceability into honey. This research encompasses a thorough examination of samples collected from the northwestern and western part of Romania, providing insights into their elemental composition. The results showed that the level of trace elements decreases in terms of traceability in honey samples (Pb was not detected in any of the analyzed samples, while Cd had a minimum content 0.001 mg/kg), ensuring its quality and health safety for consumption. Moreover, the data generated can function as a valuable resource to explore the plant’s positive eco-friendly impacts, particularly in relation to its honey.

## 1. Introduction

*Fallopia japonica* (FJ), also known as *Reynoutria japonica* (*R. japonica*), *Polygonum cuspidatum* (*P. cuspidatum)* or Japanese knotweed, is part of the Polygonaceae family [1] and is considered one of the most widely spread invasive plants in Europe [2], Australia, New Zealand and North America [3], having a negative impact on the ecosystems and communities in which it expands and, thus, representing a threat to biodiversity [4,5,6].

Originating from Asia—mostly China, Korea, Taiwan and Japan [5,7]—it was first introduced in Europe as an ornamental plant [8,9] and its spread has been escalated due to globalization and climate change [10,11,12,13,14]. 

In Romania, the plant is classified as a subspontaneous species in the riparian ecosystems of Transylvania, Maramureş, Crişana, Moldova, Banat and Oltenia, with a menacing impact on the indigenous biodiversity [15,16].

This plant proliferates in extremely diverse environments, from pastures to watercourses, forests, railways, disturbed lands and human settlements [9,14,17,18]. Its invasive character comes from its phenotypic plasticity [19] and allelopathic effects [20,21], FJ plants being able to cope with environmental changes and intoxicate the native plants through the release of some phytotoxic compounds that are able to inhibit the growth of native species [22,23,24]. This menacing character is able to produce various economic [25] and environmental negative effects in the dominated habitats of FJ, namely, regarding the regeneration capacity of the local species and associated fauna [26,27], livelihoods and human well-being [8,28], and ecological and functional modifications of the urban and rural landscapes and ecosystems [29,30,31].

Nevertheless, it has been shown that this alien species possesses a distinctive chemical composition, with pharmaceutical features [32] and an underevaluated beekeeping potential, as it represents a rich source of nectar for honeybees [17,33,34] with a high production of honey (around 200–355 kg/ha) [35,36]. The investigation performed by Ferrazzi and Marletto (1990) stands out as one of the few publications exploring the beekeeping potential of this plant based on the feeding behavior of bees [35].

Characterized as a tall herbaceous perennial species having large and profound rhizomes, FJ can measure up to 3 m high [37]. It forms dense groups of bushes that share a common rhizome structure [19] that can help them spread very quickly, together with sexual and clonal reproduction [37]. The stems are similar to bamboo sticks [38], having a woody consistency, the leaves are green and flattened, measuring 10–15 cm long, while the inflorescence contains multiple small flowers [3,39]. 

This fast-spreading plant is distinguished by its ample environmental resistance to various harsh soils and environmental conditions, especially in areas characterized by acidic soils, low nutrient levels or metal polluted habitats [40,41,42,43]. 

The recent literature has underlined the potential of this alien plant to become a natural heavy metal sink [44,45,46]. FJ has gained the interest of researchers because of its capacity to act as a green tool for the restoration of polluted areas [41,47,48,49,50], namely, its capacity to accumulate heavy metals from polluted soils where intense anthropogenic activities take place (especially industrialized, mining or non-ferrous metal extractive metallurgy areas) [51]. 

Due to growing interest for consumers in the origin and quality of food, a focus on pollution and all risks associated with the loss of biodiversity and human health has become a priority for EU policies [52]. Trace elements found in different fractions in soil, water or air represent a real risk of toxicity for any living organism, including humans, as they can enter the food chain [53,54] and produce severe dysfunctions in the human system [55,56,57,58].

Moreover, as honey is considered a natural, very nutritious product with powerful health benefits [59,60,61,62], its quality control is highly monitored by the European Commission Regulations [63,64]. It is considered that honey composition is strongly dependent on the botanical origin of harvested plants [65]; in consequence, analyzing the traceability of heavy metals from plants to honey is of great importance.

Therefore, the main objective of this study is to evaluate, thanks to samples collected from the northwestern and western parts of Romania, the presence and concentrations of macro-elements, namely, Ca, K and Mg; micro-elements, such as Cu, Fe, Mn and Se; and, most importantly, trace elements, such as Cd, Cr, Ni and Pb, in different anatomic parts of FJ plants (roots, rhizomes, stems, leaves) and their traceability into honey. 

Thus, the data generated can serve as a valuable reference for the safe consumption of FJ’s unique honey, with extraordinary bioactive properties that will be exposed in a future study. Furthermore, this research has the potential to pave the way for future advancements by providing eco-friendly solutions for managing and utilizing this plant as a beekeeping crop or raw material for biologically active substances. This, in turn, could reduce its negative environmental effects.

## 2. Results and Discussion

### 2.1. Macro-, Micro- and Trace Elements in Plant Tissues

Macro-elements including Ca, K and Mg, along with the micro-elements Cu, Fe, Mn and Se as well as trace elements such as Cd, Cr, Ni and Pb were analyzed in different anatomic parts (roots, rhizomes, stems, leaves) of FJ plants harvested from three distinct areas (Table 1, Table 2 and Table 3). 

In general, all the analyzed parts of the plant revealed significant variations in the concentration of the above-mentioned elements between the three experimental sites, namely, Merișor (Maramures County), Valea Vinului (Satu Mare County) and Bocsig (Arad County), and also across different tissues of FJ plants (post hoc Tukey HSD test, *p* < 0.01). Concerning the concentration of macro-elements in FJ plant tissues, it can be observed (please refer to Table 1) that Ca had the highest mean in all four plant tissues in the Bocsig area, with data ranging between 6047.75 mg/kg (in rhizomes) and 8991.51 mg/kg (in roots). 

Statistically significant values were also revealed for K among all areas and all FJ plant tissues (*p* < 0.01), except Valea Vinului and Bocsig, where the differences in content in leaves had no significance. The Valea Vinului area had the highest concentration of Mg and Ca in the stem parts (520.90 mg/kg and 7865.97 mg/kg, respectively), while the roots and rhizomes had no Ca detected. Comparable values were recorded in the Merișor area in terms of Ca content in the roots and rhizomes, whereas Mg showed the higher concentration in rhizomes. The fact that Ca was not detected in the roots and rhizomes in both the Merișor and Valea Vinului areas can be explained by the disturbance level of the two areas, as they are situated in proximity to former highly polluted platforms of Baia Mare (Cuprom and Romplumb). Generally, acidic soils can create calcium deficiency in plants [66]. As shown by Vidican et al. (2023), who analyzed the phytoremediation potential of this plant in the polluted area of Baia Mare (near our experimental site), four out of five tested soils were acidic soils, with a pH below 7 [50]. The existence of salinity in the soils at two of our experimental sites, Merișor and Valea Vinului, may also provide a plausible explanation for the calcium deficiency observed in both roots and rhizomes. 

In addition, the critical quality of the water and high temperatures can increase the pollution degree [67] of the area. Cordos et al. (2003) showed in their study the traceability of heavy metals in sediments in aquatic ecosystems and demonstrated the potential toxicity of the sediments [67].

Given the significant pollution in the Baia Mare area resulting from heavy metal processing, a saline solution was considered to mitigate the contamination of the soil. This approach aimed to reduce the bioaccumulation of heavy metals and prevent their potential translocation into plants. Interestingly, salinity appears to enhance the plant’s ability to withstand a mixture of heavy metals by leveraging specific physiological properties that support resistance to heavy metal stress [68]. In this context, when the concentration of NaCl in the root environment rises, there is potential for the loss of calcium from the root, leading to heightened permeability or the loss of differential permeability in the cell membranes [69].

In Table 2 are presented the values of micro-elements found in FJ plant tissues. In this regard, Cu exhibited the highest contents in the Merișor area, especially for roots (45.03 mg/kg), while the lowest concentration of Cu was observed across the other two experimental sites (Valea Vinului and Bocsig) in rhizomes and stems, where values showed no significant differences.

If Cu exceeds the limit of 20 mg/kg, in plant species, it can become toxic [70]. Thus, our results were above the limit for roots (45.03 mg/kg), rhizomes (43.60 mg/kg) and leaves in the Merișor area (23.95 mg/kg) and for leaves in the Bocsig area (23.95 mg/kg). The explanation of this surplus can be associated again with the industrialization of the area (for Merișor) and with agricultural practices (for Bocsig), as Cu is easily accumulated through the utilization of manure as fertilizer or Cu-based fungicides [71,72]. Our results are in accordance with observations reported by Sołtysiak (2020) [44] and Schmitz et al. (2023) [73], who tested the effect of Cu contamination by using different concentration treatments on FJ plants. The observations revealed a higher concentration in rhizomes [43,73]. In addition, Vidican et al. (2023) observed a surpass of Cu notably in roots and leaves of FJ plants in polluted sites in Baia Mare city [50]. However, Širka et al. (2016) observed a more notable concentration of Cu in the aboveground parts of the FJ plant (stems and leaves) from five polluted areas in Serbia [46]. Nevertheless, their mean values were lower than our results, ranging from 2.23 mg/kg in stems to 6.78 mg/kg in leaves.

As presented in Table 2, Fe had statistically significant differences (*p* < 0.01) between plant tissues across all three areas, with values ranging from 4.97 mg/kg—the lowest concentration—in Bocsig for roots to 21.06 mg/kg—the highest concentration—in Valea Vinului for stems. Very different results were observed in Serbia by Širka et al. (2016) [46] (1006.17 mg/kg—a higher average concentration of Fe in roots) and in southern Poland by Rahmonov et al. (2019) [45], with Fe levels ranging from 137 mg/kg in leaves to 6268 mg/kg in stems. Mn had significantly higher values in stem tissue from Valea Vinului (14.43 mg/kg) compared to that found in Merișor or Bocsig, where the difference lacked statistical significance. Lower values were reported in Poland by Stefanowicz et al. (2017) [74] (73 mg/kg—the upper value in the aboveground parts) nd in Serbia by Širka et al. (2016) [46] (68.91 mg/kg—the upper value in leaves).

The enormous difference in Mn content from Romania, Poland and Serbia in the same invasive plant can only be explained by taking into consideration the soil conditions (acidic soils) and the level of pollution in the experimental sites.

A higher value was also detected in Merișor for Se content in stems (0.22 mg/kg), in contrast with the rhizome value, which was four times lower 0.06 mg/kg, respectively.

The results of the current study showed significant variations in trace element (heavy metal) content within different tissues of the FJ plant, as well as across the investigated experimental sites (Table 3). Among the studied areas, Valea Vinului generally exhibited the highest concentration of heavy metals, closely followed by Merișor, and ultimately by Bocsig. The exception was for Pb, where the values were most concentrated in the Bocsig area.

Previous studies have illustrated that the extraordinary resistance of FJ to metal stress depends on the nature of the trace element involved [44] and on some environmental factors, especially soil acidity and the secondary metabolites present in the plant roots [75]. Moreover, in areas characterized by high pollution and therefore metal-rich soils, the proliferation of FJ plants is facilitated [75].

The elevated concentration of heavy metals in the Valea Vinului region compared to the Merișor area, located in proximity to the industrialized platforms of Baia Mare (approximately 17 km away), may be attributed to the presence of the Cuprom copper smelter chimney and Romplumb lead smelter chimney in Baia Mare, Romania [51]. These chimneys released toxic gases out of Baia Mare, but affect the surrounding areas.

The trace element Cd exhibited the lowest concentration in the rhizome tissue from Bocsig, measuring 0.001 mg/kg, while for the other two experimental sites, significant statistical differences (*p* < 0.01) were evident for the same plant tissue. However, both Merișor and Bocsig did not show any significant differences regarding the content of Cd in stems. The highest value for Cd was reported generally in the roots from Valea Vinului (0.694 mg/kg), followed by Merișor (0.465 mg/kg). An opposite situation was observed by Vidican et al. (2023), who reported higher values for Cd in stems and leaves [50]. Their results were in general above the values of this study.

Additionally, compared to the level of Cr outlined in the aerial parts of FJ plants by Rahmonov et al. (2019) (5.40 mg/kg) [45] and Širka et al. (2016) (2.67 mg/kg) [46], our values showed a lower concentration of Cr in leaves, ranging from 0.110 mg/kg (Bocsig area) to 1.608 mg/kg (Valea Vinului area). In fact, the FJ plant, identified as an invasive species in Europe, draws attention due to its remarkable resilience to heavy metal exposure, possibly attributable to its rhizome regeneration capability from fragments. And elements such as Cu and Cd have demonstrated the most significant influence on the morphology of this plant [76].

Overall, at a significance level of *p* < 0.01, Ni showed statistical differences in concentrations among all plant tissues, except leaves, where Merișor and Bocsig showed simillar values. Nevertheless, the most notable variances were observed between the concentration of Ni in FJ roots in Valea Vinului, with an average value of 6.295 mg/kg, and the average value of the concentration of Ni in Merișor and Bocsig (1.806 mg/kg and 1.200 mg/kg, respectively). These results were inconsistent with those obtained by Širka et al. (2016), who found higher values in FJ root and stem tissues (11.68 mg/kg and 10.12 mg/kg, respectively) [46]. In contrast, Rahmonov et al. (2019) observed a much higher concentration of Ni in leaves (36.7 mg/kg) [45].

Furthermore, as depicted in Table 3, Bocsig exhibited higher values of Pb, particularly in roots and stems, followed by Merișor, with increased values in leaf tissue. The Pb levels for the three analyzed sites highlighted a generally low concentration among all FJ plant tissues. The only exception was Merișor, which exhibited a higher value of Pb in leaves (2.084 mg/kg) and therefore, slightly exceeded the standard limits of natural Pb concentration in plants (2 mg/kg) [73]. The fact that Pb was found in higher quantity in leaves from Merișor could be explained by the long history of toxic elements processing at Baia Mare industrial platforms and also because plants primarily uptake lead from the atmosphere, predominantly via their leaves [74]. Hence, it is not surprising that the greatest concentration of Pb was found in Merișor.

On the other hand, the higher values of Pb found in Bocsig are probably the result of heavy road traffic, taking into consideration that the location of the experimental site is situated near the road, compared to the other to experimental sites, which are located far from circulated roads. Other studies did not confirm the same trend, as all of them observed values above our results and thus, beyond the allowed concentration [41,45,46,50].

When examining the levels of trace elements in the FJ plant tissues gathered across the three investigated sites, a discernible pattern emerges. This pattern is primarily influenced by the sources of the respective elements, namely, the predominant activities occurring at the collection sites. Therefore, the samples from Valea Vinului were characterized by higher values of Cd, Cr and Ni found overall in roots, rhizomes and stems, while the samples from Merișor are defined by the levels of Cr in roots and Pb in leaves. All these elements can be associated mainly with the chemical activities carried out for many decades prior to the early 2000s near Baia Mare city or to the use of fertilizers employed in agricultural practices. On the other hand, Bocsig has the higher concentration in roots, rhizomes and stems for Pb as a possible consequence of heavy road traffic or, again, as a result of the use of animal manure as fertilizer.

### 2.2. Macro-, Micro- and Trace Elements in Honey Samples

Honey contains essential nutrients and minerals that play important roles in human health [77,78]. The origin of these diverse macro-, micro- and trace elements is associated not only with the geographical origin of plants [65], but also with the influence of human activities, which may contribute to environmental pollution [79].

It is well known that honey possesses, apart from its nutritional characteristics, a wide range of pharmacological properties capable of enhancing the overall functioning of human organisms and all the natural metabolic processes associated with it [62]. However, the presence of all these elements in the human body, especially heavy metals, in excessive quantities may become detrimental and can create various health disorders [57,58].

The FJ plant is part of the same Polygonaceae family as common buckwheat, scientifically known as *Fagopyrum esculentum.* Buckwheat honey is known for its dark color, nutritional value and phenolic compounds [80]. Similar features have been obtained also for FJ honey [81], even though the literature is still scarce. Based on our knowledge, up to this moment, the only studies that have assessed FJ honey in general and its nutrients and trace elements in particular have been carried out solely in Romania [81,82]. Hence, making direct comparisons with previous studies of this unique honey variety is challenging. 

The results presented in Table 4 underline the concentrations of the same macro-, micro- and trace elements in the FJ honey samples.

The results showed that both the highest and lowest level of macro-elements were obtained in the samples collected from the same experimental area, namely, the Bocsig area. Thus, Ca exhibited a mean of 5562.75 mg/kg, while Mg showed a concentration of 406.94 mg/kg. The value for K showed no statistical differences between Merișor and Bocsig honey samples (2137.87 mg/kg and 2132.72 mg/kg, respectively), although Valea Vinului had a considerable difference (4091.25 mg/kg). These results point out the richness of important nutrients present in FJ honey, Ca, K and Mg minerals being essential for the good functioning of various biological systems: musculoskeletal, nervous and cardiac [83], among others. Furthermore, the recommended daily allowance (RDA) of these nutrients is in the range of 800–1300 mg/day for Ca, 3500 mg/day for K and 200–400 mg/day for Mg [83].

Consequently, our results are different from those presented by Bobiș et al. (2019) [81] for samples of FJ honey collected from the western part of Romania in 2019. The authors identified lower values of Ca ranging from 28.77 mg/kg to 46.09 mg/kg, while the values of K had a maximum of 6196.83 mg/kg, thus exceeding our results. As for Mg, the results of the present research were substantially higher than those exhibited by the aforementioned authors (16.27 mg/kg being the maximum concentration).

Compared to our results, lower values of the macro-elements Ca, K and Mg were observed by Pătruică et al. (2022) [82] in samples of knotweed honey from Caransebeș (western part of Romania).

The descending order of micro-elements, ranked by their concentration in the FJ honey, is as follows: Fe > Mn > Cu > Se. Copper is an essential element required for an array of biologic and metabolic processes, such as energy production, nerve conduction, immune system and fertility in women [84]. However, it can become toxic when quantities exceed the threshold limit. The RDA of Cu ranges between 1.0 and 1.6 mg per day [83].

The Cu concentration in honey samples from the Merișor area (1.29 mg/kg; *p* < 0.01) was statistically distinct from those of the other analyzed areas. The Cu concentration value in honey can be associated with the utilization of different agricultural treatments against pests that honey bees may have accumulated during their foraging [85] and transferred into honey. One of these treatments includes glyphosate, also commonly used for the eradication of FJ [86], which, in association with Cu released from industrial activities, forms strong connections that increase the toxicity level [87]. However, no significant variance in Cu level was observed between the other two experimental sites. The results were not in accordance with the observations of Pătruică et al. (2022), who found notable higher values for Caransebeș knotweed honey (4.27 mg/kg) [82]. 

Fe exhibited the highest values among all micro-elements (3.35 mg/kg) for Valea Vinului honey samples, followed by the Merișor and Bocsig areas, with values that showed important statistical differences. Fe is an important nutritional element and therefore, the consumption of FJ honey could bring important energy production to the human body [83]. The recommended intake of iron is 8–18 mg per day [83]. In contrast, its deficiency leads to anemia, as Fe is an integral component of hemoglobin and is involved in oxygen transfer [84]. 

The results obtained in our study for Fe are in line with the findings of Bobiș et al. (2019) [81], but lower compared with the findings of Pătruică et al. (2022) [82].

Levels of Mn are involved in numerous cellular enzymatic activities because it is an essential element of enzymes such as superoxide dismutase, among others [84]. The Mn values obtained in our study range from 0.969 mg/kg for the honey samples from the Bocsig area to 3.315 mg/kg for the honey samples from the Merișor area. Our results were higher than those revealed by Pătruică et al. (2022) [82]. 

The difference among the three areas concerning the Mn level can be explained by the soil composition, as these micro-elements are found naturally in soils [88]. Even so, the European Union has not established yet the maximum levels for Mn in food or in honey. 

The results exhibited for Se showed values under 1 mg/kg. Selenium is a vital constituent of selenoproteins, linked to significant reductions in the risks of various cancers. Its deficiency can potentially contribute to heart disease, hypothyroidism, and immune system deficiencies [83]. Therefore, the consumption of FJ honey, known for its elevated selenium concentration, is recommended.

The presence of trace elements (and particularly heavy metals) in the natural environment and the potential risk of their entry into the food chain pose a threat to both ecosystems and human well-being [79]. The values presented in Table 4 for all trace elements do not exceed 1 mg/kg; hence, they are in line with the European thresholds [63,89].

The largest share of harmful elements found in honey was observed for Ni (0.840 mg/kg) in the samples from the Merișor area, while Pb was not detected at all in any honey sample from all three experimental sites. Research conducted by Pătruică et al. (2022) [82] showed higher values for all discussed harmful elements, except Ni, whose values were slightly surpassed in our study. 

Generally, the most harmful trace elements found in honey are Pb and Cd. Pb, originating mainly from motor traffic and present in the air, can contaminate nectar and bee pollen directly. Conversely, Cd, sourced from the metal industry and incinerators, is transported from the soil to plants [90]. The results of the present study showed that Valea Vinului possesses the higher amount of Cd and Cr for both plant and honey samples, underlining once again that soil may be the main source for these elements.

In Romania, individuals consume approximately 0.37 kg of honey/person annually [91]. While there are no specific regulatory values for honey in the country, the European Union recommends limits of 0.1 mg/kg for Pb [89] and 0.1 mg/kg for Cd [90]. Additionally, the Joint FAO/WHO Expert Committee on Food Additives (JECFA) has set provisional tolerable weekly intake (PTWI) levels for Cd and Pb of 7 μg/kg body weight (bw) and 25 μg/kg bw, respectively [90].

The fact that in our study, FJ honey samples do not exceed the limits of harmful contaminants (heavy metals) in food is of extreme importance, mainly because our study is the first scientific certification that the FJ honey produced in the northwestern and western parts of Romania is safe for consumption.

### 2.3. Macro-, Micro- and Trace Element Traceability in FJ Plant–Honey Intractions (Translocation Factor)

With the aim of comprehending and characterizing the transfer of all the nutrients and especially of trace elements from FJ plant tissues (aboveground parts) to its honey, we analyzed the FJ plant–honey translocation factor (TF). 

Generally, the dynamics of diverse elements, particularly heavy metals, are calculated between soil and plants [50,92], soil and honey [93] or plants (inflorescence) and honey [94,95]. Moreover, taking into consideration that the research carried out before on soils near some of our investigation areas [48,50,51,93] confirmed the presence of toxic elements and the transfer of these elements into plants, the objective of our study was to assess to what degree the main nutrients as well as toxic elements are transferred from plant to honey, specifically FJ plant and honey.

As the harvest of the inflorescence part was not possible during the collecting of the other plant tissues, the transfer of macro- and micro-nutrients and especially toxic elements from plant to honey was established through a ratio between the concentration of each element found in honey samples from the three experimental sites to that in the aboveground tissues of the plants (stem and leaves). If the TF is higher than one, then a specific element from a particular experimental site has been transferred to the honey sample though the aboveground plant tissues. 

Obtaining the TF helps us to understand the dynamics of all these substances from plant to food chain, with significant emphasis on the toxic elements that can threaten human well-being if above the prescribed threshold.

The results obtained for the TF are displayed in Figure 1. Our observations of the TF indicate a strong translocation potential in the case of Mg for all three investigated areas and for Se and K in Merișor, where the data exceed the TF reference value. Conversely, the values for the other elements were below one, indicating that the transfer from the aerial parts of FJ plants to honey follows a descending trend: Mg > Se > K>Ca > Ni > Mn > Cr > Fe > Cu > Cd > Pb.

When taking into consideration the experimental sites, a higher translocation factor was noticed for Merișor, followed by Valea Vinului and Bocsig.

Macro-elements (Ca, K, Mg) exhibited a relatively high translocation factor, meaning that the existence of these elements in honey is mainly due to the plant composition.

Of all micro-elements, a noteworthy accumulation pattern was reported for Se, whose value surpassed two in the most susceptible area for pollution, namely, Merișor. Thus, the existence of this crucial micronutrient in honey is directly correlated with its existence in the plant tissues and probably in soils, as the former industrial platforms near Baia Mare city (near Merișor) were processing selenium, among other metals. 

The TF value of Mn in Merișor showed a higher value compared to the other two analyzed sites and it probably has the same above-mentioned explanation.

The calculated translocation factor for trace elements (Cd, Cr, Ni, Pb) was below 1, with values ranging from 0.002 to 0.1 for Cd, while the TF values for Cr and Ni were slightly greater. However, Pb was the only element that had no translocation factor, as it was not detected in any honey sample. 

These observations are very important, as our study demonstrates that harmful elements are not being transferred, at least not in great proportions, from the aerial plant tissues of FJ plant to its honey. 

Nevertheless, further studies have to take into consideration the impact of soil composition in order to have a full picture of the potential sources of nutrients and trace elements transferred into honey. In addition, we cannot ignore that the transmission of trace elements from plant to honey is also influenced by the honeybees’ capacity to accumulate some heavy metals into their bodies and so prevent honey contamination [96]. This aspect is not to be neglected, as the toxicity of harmful elements accumulated in honey bees bodies during foraging can affect some behavioral functions and strongly contribute to the decreases in this species’ diversity [97]. 

## 3. Materials and Methods

### 3.1. Research Location—Description

The research location for this study is the northwestern and western parts of Romania, namely, three different areas: (1) Merișor area, Maramureș county (47°39′25.2″ N 23°24′06.7″ E), in the hydrographic confluence area of the Lapus and Somes rivers, Baia Mare depression; (2) Valea Vinului area, Satu Mare county (47°43′46.9″ N 23°10′26.3″ E), on the Someș river meadow; (3) Bocsig area, Arad county (46°25′50.4″ N 21°57′47.3″ E), on the Crișul Alb river meadow (Figure 2). In all three mentioned areas, FJ forms a hedge along the rivers.

The northwestern region of Romania, in general, is renowned not only for its abundant underground mineral resources but also for being one of the country’s most environmentally compromised areas, especially the Baia Mare depression and surroundings (Maramureș and Satu Mare county). This is primarily attributed to extensive anthropogenic activities, such as mining operations, involving the extraction of valuable metals and the processing of ores containing harmful substances such as Cd, Cu, Pb and Zn (the well-known industrial platforms of Romplumb and Cuprom are examples in this case) [50,93,98]. Fortunately, these polluting activities ended in 2007, once Romania failed to uphold the environmental obligations committed to the accession to the EU. 

Two of the areas that were analyzed (Merișor and Valea Vinului) are situated in proximity to the former lead and copper production industrial platforms (Romplumb and Cuprom); hence, the occurrence of diverse trace elements in both plants and honey could be explained by this closeness. On the other hand, the Bocsig area (situated in the western part of Romania) is not related to any anthropogenic activity that could influence the existence of heavy metals in the environment.

The vegetation in these sites exhibits characteristics typical of riparian habitats, with clear signs of the invasiveness of the FJ plant, as depicted in Figure 3, and few certified beekeepers produce and commercialize FJ honey. These observations serve as the fundamental basis for choosing these locations for the current study. Concerning the coverage of FJ across the three sites, our estimates, derived from field observations, indicate that the highest FJ spread was observed in the Merișor area, encompassing approximately 80% of the total vegetation at the experimental site (with an area of around 2 ha). Following this, the Valea Vinului area showed an FJ coverage of about 60% at the experimental site, which spans approximately 1.3 ha. Lastly, in the Bocsig area, FJ covered 30% of the vegetation at the experimental site, which has an area of around 150 m^2^.

### 3.2. Plant and Honey Sampling 

The samples of FJ plants were harvested from three different counties, namely, Maramureș (1), Satu Mare (2) and Arad (3), representing the northwestern and western parts of Romania, between May and June 2023 (Figure 3). Different anatomic parts of the plant, i.e., roots, rhizomes, stems and leaves, were gathered from various locations within a single experimental site, with 3 samples taken from each site to create a representative and unified sample, ensuring precise and reliable results. The size of each collected sample from the studied experimental sites varied, with an average height ranging from 1.75 to 2 m and a weight encompassing approximately 200 to 400 g.

The samples were first washed in order to remove soil, dried for 7 days at room temperature in the dark, then ground using a laboratory mill (Grindomix^®^, GM 200—Retsch Gmbh, Haan, Germany) in order to obtain a fine powder. Then, the samples were placed in sealed bags and stored at 4 °C until testing in the Centre for Advanced Research and Extension in Apiculture (APHIS-DIA), Cluj-Napoca, Romania. 

The honey samples, three from each location, were collected directly from beekeepers, no longer than two months after the extraction, in November 2022. These samples were authenticated using palinological and sensory analyses, preserved in airtight glass containers of 1000 g and stored at a temperature of 5 °C until chemical analyses were performed. It is crucial to note that the honey samples were obtained from apiaries situated in the approximate locations where the plant samples were gathered. This meticulous alignment provides pertinent data for analysis and ensures the accuracy of the results.

### 3.3. Standards and Reagents

The following standards and reagents were used for elemental analysis: Standard materials of Mg, Ca, K, Cd, Zn, Cr, Mn, Fe, Cu, Ni, Pb and Se were provided by S.C. Cromatec Plus S.R.L. (Snagov, Romania). Nitric acid (65%) (Supelco Inc., Bellefonte, PA, USA) and hydrogen peroxide 30% (Supleco Inc., Bellefonte, PA, SUA) were also used. The stock standard solutions were prepared using acidified water (2‰ nitric acid 65% in ultrapure water (0.055 µS/cm) (*v*/*v*)). Different matrix modifiers (MerckKgaA, 6471 Darmstadt, Germany) were used in the determination: for Se, Cu and Mn, Pd + Mg (NO_3_)_2_ matrix modifier was used; for Fe, Zn and Cr, Mg(NO_3_)_2_ matrix modifier was used; and for Cd and Pb, NH_4_H_2_PO_4_ + Mg(NO_3_)_2_ matrix modifier was used. All other reagents employed for the analyses were of analytical purity. 

### 3.4. Preparation of Plant and Honey Samples for Micro- and Macro-Element Analysis with Atomic Absorption Spectrometry (AAS) 

Macro- (Ca, K, Mg), micro- (Cu, Fe, Mn, Se) and trace elements (Cd, Cr, Ni, Pb) from FJ plant and honey samples were determined using an atomic absorption spectrophotometer (AAnalyst 800 Atomic Absorption Spectrometer; PerkinElmer, Inc., Shelton, CT, USA) with atomization in the THGA graphite furnace. The method used was according to the protocol described by Pașca et al. (2017) and Socaciu et al. (2023) for plants [99,100] and Pașca et al. (2021) for honey [101].

The mineralization process of both plant and honey samples, utilizing a Berghof microwave digestion system MWS-2 calcination furnace, included a sequence of preparatory procedures. Thus, approximately 0.3 g of FJ plant (roots, rhizomes, stems, leaves) and honey homogenized samples were weighed in a Teflon digestion tube using an analytical balance. Important to mention is the fact that the honey samples, sealed in their original containers, were placed in a rotating water bath and heated at 60 °C for 30 min in order to ensure the homogeneity of each sample. A 2 mL volume of 65% nitric acid and 3 mL of 30% hydrogen peroxide were then added.

The mixture was heated in a microwave digestion system (MWS-2; Berghof, Eningen, Germany) using the following mineralization protocol: three stages were employed, with different temperatures (140 °C/180 °C/185 °C), power (60%, 70%, 80%, respectively) and various times for the digestion of the acids (10′, 9′, 1′, respectively). 

Upon completion of mineralization, the digestion vessel was unsealed following stringent safety measures and the solution was transferred into calibrated plastic containers. Then, the sample was diluted with ultrapure water to a volume of 15 mL. 

Different concentrations of standard stock solutions for the determined elements were prepared in acidified water (65% HNO_3_:H_2_O 2:1000 *v*/*v*): K 5.0 µg/L; Mg1.0 µg/L; Ca 2.0 µg/L; Cr 10.0 µg/L; Mn1.0 µg/L; Fe20.0 µg/L; Ni 50.0 µg/L; Cu 25.0 µg/L; Zn 2.0 µg/L; Cd 2.0 µg/L; Pb 50.0 µg/L and Se 100.0 µg/L. Macro-, micro- and trace elements were quantified using five-point analytical curves made from successive dilutions of the above-mentioned stock solutions. Specific wavelengths were used for constructing the calibration curve and the determination of every element: K 766.5 nm; Mg 285.2 nm; Ca 422.7 nm; Cr 357.9 nm; Mn 279.5 nm; Fe 248.3 nm; Ni 232.0 nm; Cu 324.8 nm; Zn 213.9 nm; Cd 228.8 nm; Pb 283.3 nm and Se 196.0 nm. The calibration curve variation coefficient (r^2^) for all elements was above 0.995. 

Each sample was measured in triplicate and the resulting values are presented as the mean ± standard deviation in mg/kg. All the experiments were performed at the Centre for Advanced Research and Extension in Apiculture (APHIS-DIA), Cluj-Napoca, Romania. 

### 3.5. Translocation Factor (TF)

The transfer and mobility of macro- and micro-nutrients and especially toxic elements from plant to honey can be calculated by employing a standard metric known as the translocation factor (TF). This factor was calculated using the method employed by Tomczyk et al. (2023) [95] with some modifications. Specifically, due to the absence of the plant’s inflorescence part, we determined the translocation factor (TF) by dividing the concentration of each element in honey samples from the three experimental sites by that in the aboveground tissues of the plants (stem and leaves).

### 3.6. Statistical Analysis

All data were analyzed using descriptive statistics via Statistical version 10 (developed by Stat Soft in 2010). The impact of macro-, micro- and especially trace element content on both plant parts and honey was determined through analysis of variance (ANOVA) by employing t/F-test (for single means) and post hoc Tukey HSD analysis (for correlations between group means). The results were presented as means and standard deviations. The differences were considered significant if the *p*-value was <0.01. 

## 4. Limitations of the Study

Although the study evaluated different anatomic parts of the plant (roots, rhizomes, stems, leaves), there is a part that was not taken into consideration in the study, namely, the inflorescence. Because the plant samples were collected between May and June, the inflorescences were not yet available, as the blooming period for FJ in Romania is in September. However, the results have shown that honey receives a small amount of trace elements compared to all the analyzed parts of the plant. We can assume that the missing inflorescence part did not have a great concentration of heavy metals; otherwise, bees, as bio-indicators, could have had accumulated these elements in their bodies or transferred them to the honey. As numerous studies have shown, bees have the capacity to accumulate certain toxic elements present in plants, soil and atmospheric air in their tissues or transfer them to bee-derived products [102,103,104,105], which are often more tainted with toxic substances than the plants from which they are gathered [106]. Another factor contributing to the rise in concentrations of toxic elements in the bee’s body occurs during the transformation of raw honey into the final product [107]. Whilst many metals can be transferred to honeybees, three of the most common are Cd, Cu and Pb [108,109], with destructive consequences for bee colonies. Despite the constraints, the current study is just a part of a more complex research that has the goal of assessing FJ honey’s bioactive properties. Thereby, in our future manuscripts, a comprehensive analysis, including the inflorescence part, will be performed to give a thorough evaluation of all elements’ traceability in honey. This will avoid an incomplete understanding of the pathways through which macro-, micro- and trace elements enter the honey composition, hindering the accurate assessment of its safety and quality. 

The limited number of beekeepers that are aware of the beekeeping potential of this invasive plant could also influence the present study, as the number of samples from the analyzed areas were limited. Furthermore, in other regions of Romania where the FJ plant is prevalent, there is a lack of available information regarding the production of FJ honey, as well as insights into beekeepers’ awareness of the beekeeping potential associated with this plant. Hopefully, all the observations gathered in this study will provide novel data that could be employed by beekeepers as a valuable reference for utilizing this plant to explore the plant’s positive eco-friendly impacts, particularly in relation to its honey.

## 5. Conclusions 

The present study analyzed samples of FJ plants and honey from three different experimental sites located in the northwestern and western parts of Romania in order to assess the concentration of macro-, micro-, and harmful elements. Among all three experimental sites, Valea Vinului exhibited, in general, the highest contents of elements, followed by Merișor and lastly, by Bocsig. The exception was for Ca and Pb, where the values were more elevated in Bocsig.

The results have confirmed the abundance of crucial macro-nutrients in FJ plants, namely, Ca, K and Mg. Micro-elements with higher concentrations for Cu, especially in roots (Merișor area), and Fe in aboveground parts (Valea Vinului area). Regarding trace elements found in FJ plants, Ni exhibited the highest values of all tested harmful elements. Comparatively, Pb showed the lowest content in all samples, with one exception (leaves from Merișor).

This research also examined the concentration of all these elements in FJ honey, revealing important concentrations of Ca, K and Mg, while the content of harmful elements in honey was within the limits established by the European Community for suitable consumption—all trace elements were below 1 mg/kg, with Pb undetected in any honey sample. 

This study also employed the translocation factor (TF), indicating the extent to which these elements have been translocated from the plant’s aboveground parts into the honey. 

The descending order of element translocation was as follows: Mg > Se > K > Ca > Ni > Mn > Cr > Fe > Cu > Cd > Pb. 

Taking into consideration the above-mentioned aspects, based on our knowledge, this is the first scientific study to compare the essential minerals and trace elements in FJ plants to those of FJ honey. It is also the first certification of the fact that FJ honey produced in the northwestern and western parts of Romania does not exceed the limits of harmful contaminants in food, making it a variety of honey adhering to safety standards and promoting the health of consumers.

## Figures and Tables

**Figure 1 plants-13-00428-f001:**
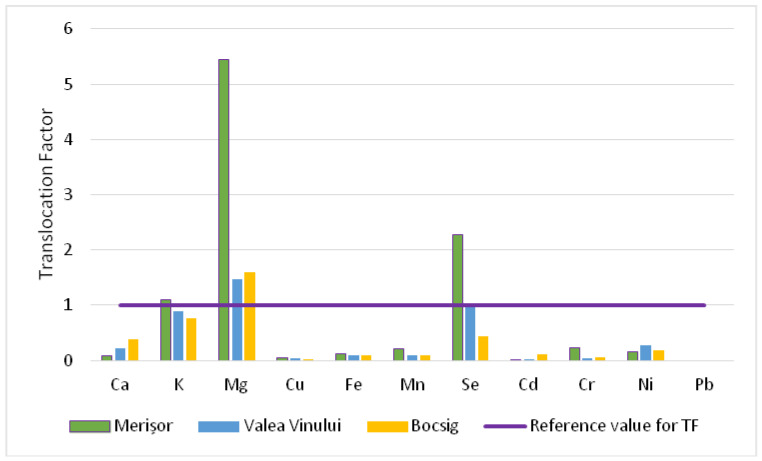
The translocation of macro-, micro- and trace elements found in FJ plants to their honey in the three experimental sites.

**Figure 2 plants-13-00428-f002:**
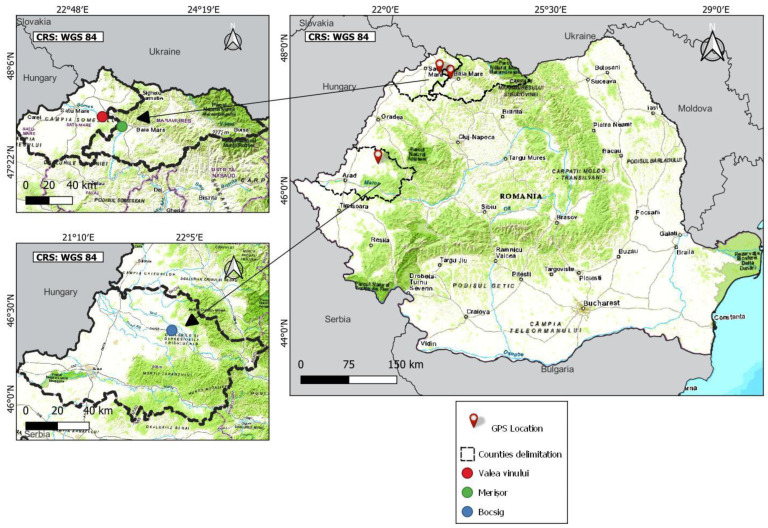
Location of the experimental sites (QGIS VERSION 3.28.7).

**Figure 3 plants-13-00428-f003:**
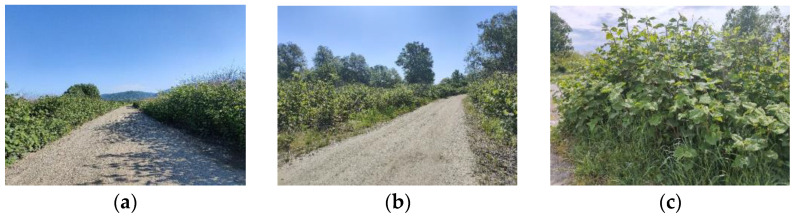
Location of the study sites. (**a**) Merișor area; (**b**) Valea Vinului area; (**c**) Bocsig area. (Source: personal collection of Alexandra-Antonia Cucu).

**Table 1 plants-13-00428-t001:** The concentration of macro-elements in FJ plant tissues (mg/kg).

Indicator (Elements)	Samples (Plant Tissues)	Experimental Site		
		Merișor	Valea Vinului	Bocsig
Ca	Roots	nd	nd	8991.51 ± 59.904 ^a^
	Rhizomes	nd	nd	6047.75 ± 109.273 ^a^
	Stems	6587.67 ± 624.125 ^b^	7865.97 ± 13.801 ^a^	7704.45 ± 35.214 ^a^
	Leaves	2361.56 ± 80.063 ^c^	6614.36 ± 20.588 ^b^	6764.95 ± 59.805 ^a^
K	Roots	3478.21 ± 26.412 ^b^	3935.88 ± 20.588 ^a^	1125.99 ± 27.133 ^c^
	Rhizomes	3817.59 ± 24.691 ^a^	3628.35 ± 5.628 ^b^	1265.45 ± 5.922 ^c^
	Stems	1703.20 ± 171.017 ^b^	3038.79 ± 15.526 ^a^	1273.65 ± 39.792 ^c^
	Leaves	2522.72 ± 35.158 ^a^	1569.88 ± 153.041 ^b^	1501.49 ± 23.218 ^b^
Mg	Roots	195.76 ± 4.726 ^b^	273.26 ± 5.060 ^a^	129.64 ± 0.810 ^c^
	Rhizomes	520.53 ± 1.058 ^a^	355.50 ± 4.538 ^b^	130.76 ± 4.018 ^c^
	Stems	143.99 ± 4.478 ^b^	520.90 ± 1.725 ^a^	93.85 ± 4.894 ^c^
	Leaves	nd	140.37 ± 0.514 ^b^	160.77 ± 5.523 ^a^

Values are given as an average of three replications ± standard deviations. Values in the same row with different non-capital letters a, b, c are statistically significantly different between groups (post hoc Tukey HSD test, *p* < 0.01). Detection limit (LOD) and quantification limit (LOQ) were determined for macro-elements as follows: LOD for Ca: 0.03 µg/L; LOQ for Ca: 0.01 µg/L; LOD for K: 0.005 µg/L; LOQ for K: 0.015 µg/L; LOD for Mg: 0.012 µg/L; LOQ for Mg: 0.004 µg/L. “nd”—not detected (values below the detection limit (LOD)).

**Table 2 plants-13-00428-t002:** The concentration of micro-elements in FJ plant tissues (mg/kg).

Indicator (Elements)	Samples (Plant Tissues)	Experimental Site		
		Merișor	Valea Vinului	Bocsig
Cu	Roots	45.03 ± 0.223 ^a^	25.76 ± 0.189 ^b^	17.05 ± 0.015 ^c^
	Rhizomes	43.60 ± 0.375 ^a^	5.99 ± 0.051 ^b^	5.76 ± 0.017 ^b^
	Stems	8.69 ± 0.007 ^a^	7.86 ± 0.098 ^b^	7.92 ± 0.008 ^b^
	Leaves	23.95 ± 0.059 ^a^	10.13 ± 0.014 ^b^	23.27 ± 0.017 ^a^
Fe	Roots	16.38 ± 0.020 ^a^	14.16 ± 0.006 ^b^	4.97 ± 0.010 ^c^
	Rhizomes	17.48 ± 0.015 ^a^	16.44 ± 0.003 ^b^	12.57 ± 0.022 ^c^
	Stems	14.29 ± 0.088 ^b^	21.06 ± 0.006 ^a^	12.07 ± 0.010 ^c^
	Leaves	4.99 ± 0.034 ^c^	15.72 ± 0.065 ^a^	10.76 ± 0.014 ^b^
Mn	Roots	10.14 ± 0.069 ^a^	8.68 ± 0.100 ^b^	7.06 ± 0.077 ^c^
	Rhizomes	4.25 ± 0.039 ^b^	10.44 ± 0.116 ^a^	1.42 ± 0.032 ^c^
	Stems	3.56 ± 0.024 ^b^	14.43 ± 0.186 ^a^	3.81 ± 0.055 ^b^
	Leaves	11.43 ± 0.027 ^a^	8.14 ± 0.039 ^b^	6.704 ± 0.017 ^c^
Se	Roots	0.17 ± 0.002 ^b^	0.21 ± 0.000 ^a^	0.10 ± 0.000 ^c^
	Rhizomes	0.06 ± 0.000 ^b^	0.15 ± 0.000 ^a^	0.12 ± 0.000 ^a^
	Stems	0.22 ± 0.000 ^a^	0.11 ± 0.002 ^b^	0.07 ± 0.000 ^b^
	Leaves	0.12 ± 0.000 ^a^	0.11 ± 0.002 ^a^	0.11 ± 0.001 ^a^

Values are given as an average of three replications ± standard deviations. Values in the same row with different non-capital letters a, b, c are statistically significantly different between groups (post hoc Tukey HSD test, *p* < 0.01). Detection limit (LOD) and quantification limit (LOQ) were determined for micro-elements as follows: LOD for Cu: 0.042 µg/L; LOQ for Cu: 0.014 µg/L; LOD for Fe: 0.180 µg/L; LOQ for Fe: 0.06 µg/L; LOD for Mn: 0.015 µg/L; LOQ for Mn: 0.005 µg/L; LOD for Se: 0.15 µg/L; LOQ for Se: 0.05 µg/L.

**Table 3 plants-13-00428-t003:** The concentration of trace elements in FJ plant tissues (mg/kg).

Indicator (Elements)	Samples (Plant Tissues)	Experimental Site		
		Merișor	Valea Vinului	Bocsig
Cd	Roots	0.465 ± 0.017 ^b^	0.694 ± 0.021 ^a^	0.029 ± 0.000 ^c^
	Rhizomes	0.152 ± 0.003 ^b^	0.401 ± 0.002 ^a^	0.001 ± 0.000 ^c^
	Stems	0.012 ± 0.000 ^b^	0.450 ± 0.048 ^a^	0.003 ± 0.000 ^b^
	Leaves	0.124 ± 0.000 ^a^	0.020 ± 0.000 ^b^	0.004 ± 0.000 ^c^
Cr	Roots	1.587 ± 0.035 ^a^	1.177 ± 0.022 ^b^	0.929 ± 0.006 ^c^
	Rhizomes	0.146 ± 0.000 ^c^	1.586 ± 0.042 ^a^	0.956 ± 0.001 ^b^
	Stems	0.093 ± 0.000 ^c^	1.283 ± 0.013 ^a^	0.755 ± 0.001 ^b^
	Leaves	0.110 ± 0.000 ^c^	1.608 ± 0.008 ^a^	1.009 ± 0.000 ^b^
Ni	Roots	1.806 ± 0.008 ^b^	6.295 ± 0.032 ^a^	1.200 ± 0.000 ^c^
	Rhizomes	0.695 ± 0.006 ^b^	1.875 ± 0.021 ^a^	0.134 ± 0.000 ^c^
	Stems	0.617 ± 0.003 ^b^	2.745 ± 0.011 ^a^	0.275 ± 0.001^c^
	Leaves	1.031 ± 0.004 ^a^	0.521 ± 0.000 ^c^	1.060 ± 0.003 ^a^
Pb	Roots	0.006 ± 0.001 ^c^	0.042 ± 0.003 ^b^	0.216 ± 0.016 ^a^
	Rhizomes	nd	0.014 ± 0.004 ^a^	0.011 ± 0.003 ^a^
	Stems	0.001 ± 0.000 ^b^	0.003 ± 0.002 ^b^	0.044 ± 0.004 ^a^
	Leaves	2.084 ± 0.008 ^a^	0.051 ± 0.002 ^c^	0.124 ± 0.004 ^b^

Values are given as an average of three replications ± standard deviations. Values in the same row with different non-capital letters a, b, c are statistically significantly different between groups (post hoc Tukey HSD test, *p* < 0.01). Detection limit (LOD) and quantification limit (LOQ) were determined for trace elements as follows: LOD for Cd: 0.006 µg/L; LOQ for Cd: 0.002 µg/L; LOD for Cr: 0.012 µg/L; LOQ for Cr: 0.004 µg/L; LOD for Ni: 0.210 µg/L; LOQ for Ni: 0.07 µg/L; LOD for Pb: 0.150 µg/L; LOQ for Pb: 0.05 µg/L. nd—not detected (values are below the detection limit (LOD)).

**Table 4 plants-13-00428-t004:** The concentration of macro-, micro- and trace elements in FJ honey (mg/kg).

Element	Concentration in Honey Samples from the Three Experimental Sites		
	Merișor (n = 3)	Valea Vinului (n = 3)	Bocsig (n = 3)
Macro-elements			
Ca	674.45 ± 0.702 ^c^	3281.39 ± 0.891 ^b^	5562.75 ± 0.971^a^
K	2132.72 ± 0.663 ^b^	4091.25 ± 0.911^a^	2137.87 ± 0.833 ^b^
Mg	785.35 ± 0.682 ^b^	972.44 ± 0.801^a^	406.94 ± 0.642 ^c^
Micro-elements			
Cu	1.29 ± 0.0166 ^a^	0.76 ± 0.025 ^b^	0.78 ± 0.006 ^b^
Fe	2.28 ± 0.097 ^b^	3.35 ± 0.045 ^a^	1.91 ± 0.011 ^c^
Mn	3.13 ± 0.010 ^a^	2.19 ± 0.072 ^b^	0.97 ± 0.022 ^c^
Se	0.77 ± 0.021 ^a^	0.22 ± 0.031 ^b^	0.08 ± 0.002 ^c^
Trace elements			
Cd	0.001 ± 0.000 ^a^	0.001 ± 0.000 ^a^	0.001 ± 0.000 ^a^
Cr	0.044 ± 0.002 ^c^	0.103 ± 0.000 ^a^	0.090 ± 0.001 ^b^
Ni	0.840 ± 0.006 ^a^	0.269 ± 0.001^b^	0.235 ± 0.009 ^c^
Pb	nd	nd	nd

Values are given as an average of three replications ± standard deviations. Values in the same row with different non-capital letters a, b, c are statistically significantly different between groups (post hoc Tukey HSD test, *p* < 0.01). Detection limit (LOD) and quantification limit (LOQ) were determined for the elements in honey as follows: LOD for Ca: 0.03 µg/L; LOQ for Ca: 0.01 µg/L; LOD for K: 0.005 µg/L; LOQ for K: 0.015 µg/L; LOD for Mg: 0.012 µg/L; LOQ for Mg: 0.004 µg/L; LOD for Cu: 0.042 µg/L; LOQ for Cu: 0.014 µg/L; LOD for Fe: 0.180 µg/L; LOQ for Fe: 0.06 µg/L; LOD for Mn: 0.015 µg/L; LOQ for Mn: 0.005 µg/L; LOD for Se: 0.15 µg/L; LOQ for Se: 0.05 µg/L. LOD for Cd: 0.006 µg/L; LOQ for Cd: 0.002 µg/L; LOD for Cr: 0.012 µg/L; LOQ for Cr: 0.004 µg/L; LOD for Ni: 0.210 µg/L; LOQ for Ni: 0.07 µg/L; LOD for Pb: 0.150 µg/L; LOQ for Pb: 0.05 µg/L. nd—not detected (values are below the detection limit (LOD)).

## Data Availability

Data are contained within the article.

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
