# Peer review of "Evaluation of the Main Macro-, Micro- and Trace Elements Found in Fallopia japonica Plants and Their Traceability in Its Honey: A Case Study from the Northwestern and Western Part of Romania"

_plants, 2024, doi:10.3390/plants13030428_

Round 1

Reviewer 1 Report

Comments and Suggestions for Authors

GENERAL COMMENTS: 

The authors present the results of the elemental composition (macro-, micro- and trace elements) found in Fallopia Japonica plant and their traceability in the related honeys from the North-Western and Western part of Romania. Unfortunately, however, no specific explanations for particular results were provided. Therefore, I have made my recommendation on the above basis.

1.     The abbreviation FJ was written in italic in the abstract, but not in the rest of the text; please correct this. 2.     Line 457 – „manganese (Mg)“ should be „magnesium (Mg)“ 3.     Please explain  which matrix modifier was used. 4.     Please explain the meaning of "Standard solutions were prepared using acidified water as a solvent" and the values in brackets for the elements listed. 5.     Add instrumental parameters for measurement conditions.

6.     Line 468 – „…from 467 FJ plant and honey samples was determined…“should be „…from 467 FJ plant and honey samples were determined…“

7.     Throughout the manuscripts, the authors have repeated the names of the elements and symbols many times. Please, use only one spelling. 8.     What type of mill did the authors use to prepare the samples? Did the authors check for possible contamination from the mill? 9.     The authors collected plant material between May-June 2023 and honey samples in November 2022. I think this is the main problem of this work. The authors have given a translocation factor for plant honey, but how can you conclude this if you did not use the honey from the plants studied! Please explain. In addition, the authors mentioned that the soils near their study areas have already been analysed, and the translocation factor has also already been determined - please explain the novelty of this work. 10.  A better explanation of the results obtained in the case of the translocation factor is required. For example, the authors concluded that a relatively high TF in macroelements is mainly due to plant composition, but what about soil composition? Please also indicate how the authors calculated the TF if they used the results for leaves and stems. 11.  Table 4- please correct  

12.  I think that the authors should rewrite the conclusion. It is not necessary to explain again the properties of the plant. Only the main conclusions from the results obtained should be included in the conclusion. Please correct this part accordingly.

13.  Some grammar and format error should be carefully revised in the manuscript.

14.  Please check the way of writing references. Standardize the writing of references in the reference section. 

Comments on the Quality of English Language

1.     Some grammar and format error should be carefully revised in the manuscript.

Author Response

We thank you for the disponibility to revise our manuscript and for the suggestions you have made to improve the manuscript. Please find attached the point-by-point answers to your comments.

Thank you

Reviewer 2 Report

Comments and Suggestions for Authors

Revision of manuscript: Evaluation of the main macro-, micro-, and trace elements found in Fallopia Japonica plant and their traceability into its related honey: a case study from the North-Western and Western part of Romania (plants-2797404)

The topic of the article is interesting. The main goal of the manuscript is to investigate how the different metal contamination of the Fallopia japonica plant influences the content of trace elements in honey. But I think it needs to be improved, especially regarding the presentation of the data and some information on the analytical part, facilitating reading and expanding the audience.
My recommendation is that the authors make a minor improvement on the manuscript and resubmit it for review.

Title: please do not use capital letters Japonica

In tables 1-4, and throughout the manuscript, please pay attention to significant figures

Tab.1: It seems very unlikely that traces of calcium are not found in roots and rhizomes and the explanation seems to be inconsistent.

Furthermore, no data relating to the LOD and LOQ of the analytical method were provided

Similarly, in Table 4 lead is defined as undetected but no information was provided on LOD and LOQ.

Line 168:

the high concentration variability of all the elements analyzed can certainly be attributed to the environmental contamination of the areas under study. Therefore, some elements are more present in some samples and some plant parts (roots rather than leaves and vice versa). However, the small number of samples makes this result less significant

Line 338: in the cited reference a maximum limit of 0.10 mg/kg is set for lead and no maximum limit is established for cadmium, please correct this sentence Line 442: it is unclear why a pool of the three samples was created instead of analyzing the individual samples collected

Line 452: The Authors affirm “the honey samples were obtained from apiaries situated precisely in the same locations where the plant samples were gathered” but no confirmatory melissopalynological analysis is reported. No one assures us that the bees chose the FJ pollen.

Author Response

(The authors gave the same response as above.)

Reviewer 3 Report

Comments and Suggestions for Authors

Manuscript Comments

Title: Evaluation of the main macro-, micro-, and trace elements found 2 in Fallopia Japonica plant and their traceability into its related honey: a case study from the North-Western and Western part  of Romania

The manuscript submitted by Alexandra-Antonia Cucu et al. provides detailed information on the concentration of macro-elements, namely calcium (Ca), potassium (K), magnesium (Mg), micro-elements, such as copper (Cu), iron (Fe), manganese (Mn), selenium (Se) or trace elements: cadmium (Cd), chromium (Cr), nickel (Ni), and lead (Pb), in different anatomic parts of Fallopia japonica (FJ) plant (roots, rhizomes, stems, leaves) and in honey, from each location, where the plants were collected (North-Western and Western part of Romania).

Such a study is essential in understanding plant metal translocation. In my opinion, this article provides vital information for researchers dealing with environmental and health risk assessments. The paper falls within the Aims and Scope of the journal. The cited literature is up-to-date. This article contains new aspects and is of general interest. The abstract covers the information presented in the manuscript, and the keywords suit its content. The paper has an appropriate structure. I generally find this work interesting; however, I have a suggestion to improve the manuscript.

In my opinion, the conclusion: “Furthermore, the data generated can serve as a  valuable reference for the use of this plant towards sustainable purposes such as beekeeping.” should be rewritten. In my opinion, sustainable purposes are inappropriate for alien species.

In the “Materials and Methods” section, the authors should mention how big one sample has been considered.  It should be more precisely specified, and from how large an area was collected.

What about the certified reference material in elements estimation?

It is a pity that the Authors did not study the inflorescences, although, in the Conclusions and Limitations of the study, we find an explanation of this fact.  In the same chapter, we read “We can assume that the missing inflorescence part hadn’t had a great concentration of heavy metals; otherwise, bees, as bioindicators, could have accumulated these elements in their bodies”,  but trace metal accumulation in bees has not been studied. It would be worthwhile to supplement with data from the literature or other studies.

In my opinion, the chapter Conclusions and Limitations of the study should be reworded. First, the limitations should be indicated, and then the conclusions. It is not apparent how important and valuable the conclusions are, even though inflorescences were not included in the study.

In my opinion, the text of the whole manuscript should be checked by a Native Speaker.

Author Response

(The authors gave the same response as above.)
